# Snake Antivenoms—Toward Better Understanding of the Administration Route

**DOI:** 10.3390/toxins15060398

**Published:** 2023-06-15

**Authors:** Erika Gamulin, Sanja Mateljak Lukačević, Beata Halassy, Tihana Kurtović

**Affiliations:** Centre for Research and Knowledge Transfer in Biotechnology, University of Zagreb, Rockefellerova 10, 10000 Zagreb, Croatia; egamulin@unizg.hr (E.G.); sanjamalu@gmail.com (S.M.L.); bhalassy@unizg.hr (B.H.)

**Keywords:** antivenom, passive immunotherapy, administration route, envenoming treatment, snakebite, venom

## Abstract

Envenomations induced by animal bites and stings constitute a significant public health burden. Even though a standardized protocol does not exist, parenterally administered polyclonal antivenoms remain the mainstay in snakebite therapy. There is a prevailing opinion that their application by the *i.m.* route has poor efficacy and that *i.v.* administration should preferentially be chosen in order to achieve better accomplishment of the antivenom therapeutic activity. Recently, it has been demonstrated that neutralization not only in the systemic circulation but also in the lymphatic system might be of great importance for the clinical outcome since it represents another relevant body compartment through which the absorption of the venom components occurs. In this review, the present-day and summarized knowledge of the laboratory and clinical findings on the *i.v.* and *i.m.* routes of antivenom administration is provided, with a special emphasis on the contribution of the lymphatic system to the process of venom elimination. Until now, antivenom-mediated neutralization has not yet been discussed in the context of the synergistic action of both blood and lymph. A current viewpoint might help to improve the comprehension of the venom/antivenom pharmacokinetics and the optimal approach for drug application. There is a great need for additional dependable, practical, well-designed studies, as well as more practice-related experience reports. As a result, opportunities for resolving long-standing disputes over choosing one therapeutic principle over another might be created, improving the safety and effectiveness of snakebite management.

## 1. Overview

Venoms evolved as a valuable adaptive trait that played a vital role in the easier survival and reproductive success of various venomous species [1]. Venomous animals possess specialized exocrine glands and apparatuses for the production of venom and its active delivery into the victim’s body with the aim of predation, self-defense or intraspecific competition [2,3,4]. Among more than 100,000 venomous species in the world, in most parts, snakes have been considered the most important medically on account of the frequency of their bites as the main cause of human envenoming [5]. Their venoms comprise a variety of more than a hundred different pharmacologically active compounds capable of triggering a wide range of serious and often life-threatening pathophysiological manifestations [6,7,8]. It has been estimated that over 2.7 million people suffer from the consequences of envenomation annually, with fatalities ranging from 81,000 to 138,000, while in the case of survival, more than 400,000 remain maimed for life [8,9,10]. As such, snakebite envenoming constitutes a significant public health burden particularly affecting poor and densely populated rural tropical regions [7,11,12]. In 2017, after decades of inattention, the WHO developed a comprehensive strategy with the goal to reduce its devastating impact through the assurance of the global availability of safe and effective antivenoms [10], specific and validated life-saving therapeutics capable of neutralizing and reversing the lethal and tissue-damaging toxic effects of venom components [5]. Nowadays, the whole world faces a critical and longstanding shortage of antivenoms, affecting, in the first place, developing countries [13,14,15,16] but high-income countries as well [17,18,19], whose alleviation aims for the development of feasible and profitable production strategies, rational use of medications and implementation of well-designed treatment protocols [10,13,20].

The active compounds of antivenoms are either whole or enzyme-digested equine, sometimes ovine immunoglobulins of the G class (IgGs) raised against venom from a single or several medically relevant snake species [21]. Their production began more than a century ago when the way for passive serotherapy was paved [22], and since then, it has been modified in line with technological innovations and Good Manufacturing Practice requests [23,24], although advances toward more effective and safe products are still needed [25]. Alternative approaches to conventional antivenoms include monoclonal antibodies, aimed at targeting the most relevant toxins [26], oligonucleotide aptamers [27], nanoparticles [28,29], peptides [30] and small molecule inhibitors [31]. Although next-generation therapeutics also have the proven preclinical ability of neutralizing the venom components of interest [32], none have reached the clinical trial level yet [24]. Therefore, traditional antivenoms for now remain the mainstay in snakebite envenoming treatment. In general, they are well suited for their purpose as long as they comply with the conditions of safety for human use and efficacy in abolishing the venom’s action [33]. Safety is guided by the manufacturing procedure’s conditions, affecting the purity, physicochemical characteristics and stability of the preparation [34]. Efficacy, a measure of neutralizing potency [33], principally relies on specificity [35] and the concentration of the antibodies [13]. Moreover, it depends on the well-timed availability of a sufficient amount of antivenom within tissue compartment(s) in which target molecules are present, preferentially bypassing activation of the patient’s immunological response. All current antivenoms are based either on whole IgG molecules or their antigen-binding domains in the form of F(ab’)_2_ and Fab fragments [36]. Accordingly, due to the variable molecular mass of the active compound, they exhibit distinct pharmacokinetic profiles [35]. Consequently, the competencies of IgGs or their fragments to achieve successful detoxification outcomes for the most part depend on their ability to find themselves fast enough and at a high concentration in a common distribution space together with venom toxins. Namely, the common distribution space is a site where the capture, extraction or redistribution and, finally, elimination should occur, ideally before the manifestation of deleterious effects takes place [37,38]. In other words, venom–antivenom binding should be facilitated during or even prior to the delivery of venom components from the bite site to the place of action [36], as once envenomation symptoms are established, diminished efficacy could be observed [39,40,41]. Furthermore, the pharmacokinetic properties of IgG-, F(ab’)_2_- and Fab-based antivenoms are not always compatible with those of the venom of interest, and sometimes an extensive mismatch in their pharmacokinetic behavior occurs. Therefore, the selection of the optimal antivenom therapy requires an accurate evaluation of both venom toxicokinetics and antivenom pharmacokinetics in order to establish an adequate therapeutic dose and injection route [42].

Preclinical [43] and clinical studies [44,45] showed that Fab antivenoms have a much larger volume of distribution, compared to those composed of F(ab’)_2_ fragments or IgGs, due to their low molecular mass which enables them to readily reach the extravascular compartment [46] and redistribute venom antigens to the vascular space [47]. For the same reason, the decline in their concentration occurs more rapidly, mostly via renal filtration, with the elimination half-life between only 4 and 24 h [46,48]. F(ab’)_2_ fragments and IgGs, due to their higher molecular weight, persist in the circulation for a longer period of time before being removed, showing a prolonged elimination half-life that spans between 2 and 4 days [45]. In addition, they possess two antigen-binding sites compared to monovalent Fab fragments, enabling them to form large, stable multivalent immuno-complexes with toxins that are eliminated dominantly by phagocytic cells in the reticuloendothelial system [35]. Based on different pharmacokinetic features, the optimal treatment of venom-induced pathophysiological effects requires the most appropriate antivenom format. Fab antivenoms are considered more suitable for elapid venoms, abundant with low-molecular mass toxins, while F(ab’)_2_ and IgG formulations exhibit properties more effective in counteracting larger molecules characteristic of viperids [33]. However, even if the antivenom and venom remain in the central compartment equally long, discrepancies from the ideal scenario could possibly occur, as demonstrated in studies reporting that after transient improvement, the signs of recurrence appeared [36,46]. Such a phenomenon is associated with the redistribution of the venom into the circulation by slow continuous absorption from a depot site following the elimination of the circulating antivenom or by reversible venom–antivenom binding [49]. Consequently, repeated administration of antivenom is needed to maintain the therapeutic level of its active compound. This is primarily characteristic of Fab preparations, due to their high clearance rate from the vascular space and their absence during the late phase of envenomation when the reappearance of non-neutralized toxins occurs [50]. IgG or F(ab’)_2_ antivenoms persist in the circulation for a longer period of time, therefore ensuring the presence of neutralizing antibodies in sufficiently high concentrations for the complete abolishment of the circulating venom’s activity [33].

Other than antibody specificity and concentration, as well as the composition profile, antivenom efficacy might be highly influenced by the route of administration as well. There is no standardized protocol for antivenom administration across Europe, similar to many other regions of the world [51]. It is a WHO recommendation [52] that, whenever possible, snake antivenoms should be given intravenously (*i.v*.) due to the higher speed of distribution and greater bioavailability of neutralizing antibodies in comparison to other routes. Slow *i.v.* infusion over 30–45 min allows the cessation of antivenom administration if immediate adverse reactions develop [53]. *I.v.* administration is logistically more demanding as it must be performed under the close supervision of health care professionals within medical facilities. Their accessibility is often hindered by the remoteness of the snakebite-prone areas leading to the delayed transportation and, consequently, treatment of victims, which ultimately reduces the chances of a successful therapy outcome [8,54,55,56]. Intramuscular (*i.m.*) administration brings a notably lower risk of antivenom-associated side effects and is easier to give in resource-poor or remote settings in the absence of expert medical aid [52]. However, there is a prevailing opinion that the *i.m.* route is less effective and leads to lower bioavailability, a longer time to reach the maximum concentration and a delayed and incomplete neutralization of toxins [35]. Blood levels never reach those rapidly achieved by *i.v.* application. Therefore, the WHO advises the *i.m.* route as an alternative approach at peripheral first aid stations far from medical care, as well as in the case when *i.v.* access has proven to be impossible [52]. Local administration of antivenom at the site of the bite should not be performed, as it is extremely painful and may increase intracompartmental pressure [52]. Accordingly, the majority of commercially available antivenoms are intended for *i.v.* infusion. The exceptions are those aimed for the treatment of other venomous animals’ bites/stings [57,58], which are either consistently given *i.m.*, despite the increasing concerns of their lower effectiveness when applied by this path [59], or by both the *i.v.* and *i.m.* routes, since it is still unclear which one is more effective [60]. In the case of snakebites, *i.v.* administration represents the method of choice whenever professional medical care is available [58,61,62]. However, there is also a significant number of manufacturers whose products are still prescribed for *i.m.* and/or *s.c.* application [63,64,65,66,67,68,69,70]. It might represent a not-so-incomprehensible concept if snake venoms are anticipated as complex mixtures of proteins with variable molecular mass that are, in most envenomation cases, injected into the interstitial space either by the *s.c.* or *i.m.* route [9] and whose absorption into the bloodstream may occur by the way of blood capillaries or small lymphatic vessels, depending on their size [71]. Consequently, venom components exhibit different toxicokinetic profiles [62]. On the other hand, antivenoms with a uniform composition, which involves only large molecules, if given by the same route, reach the central compartment by slow diffusion into the initial lymphatics [62,72]. In addition, there is proof that the lymphatic system not only plays a role in venom distribution and bioavailability [73] but also serves as a compartment where antivenom, extravasated from the blood after *i.v.* administration, eliminates a substantial amount of toxins before lymph reaches the systemic circulation [74]. For now, there are no cognitions about *i.m.* antivenom-mediated neutralization within the lymphatics.

It was our intention to summarize the laboratory and clinical findings on the *i.m.* and *i.v.* routes of antivenom administration, as two different therapeutic approaches with distinct pharmacokinetic properties and implications for the pharmacodynamics accordingly, especially in light of new cognitions from the field. The experts should be aware that there is still, even after many decades, the need for additional well-designed, pragmatic and reliable studies but also much more reports on experiences from the practice. So, opportunities for the resolution of the established controversies associated with the preference of one therapeutic principle over the other, contributing to the safety and efficacy of snakebite management, might be created. In addition, since a wider range of innovations to traditional antivenoms is now being developed, the emergence of new-generation therapeutics, which will likely have different characteristics, could be expected [75]. An up-to-date perspective on the knowledge gained so far could possibly contribute to a better understanding of their pharmacokinetics and the optimal administration route as well, ensuring the fulfillment of fit-to-purpose conditions.

## 2. Preclinical Studies

The intramuscular (*i.m.*) route is a parental type of drug administration via a syringe or a needle into body tissue whereupon the drug diffuses from the muscle into the surrounding interstitial fluid and finally into the blood. Preparations for *i.m.* use are commonly injected into gluteal or deltoid muscle of which the second one has been advised as the preferential choice within clinical practice due to higher blood flow [76,77]. As for the other administration principles, the pros and cons associated with the *i.m.* route have been observed. It allows a rapid absorption of specific medications into the circulation and their well-timed onset of action [78] and is considered highly effective during emergency scenarios [79]. On the other hand, medical incidents such as local area trauma and pain caused by sharp injection needle and tension from the drug volume, aseptic inflammatory reaction down the developed muscle channel, nerve damage and infection might occur [80].

Intravenous (*i.v.*) administration is the fastest and the most reliable way of releasing a drug into the circulatory system with the immediate delivery of a possible large fluid volume [76,81]. Except for complete drug availability, it enables, by the control of the administration rate, constant plasma concentrations at the required level [82]. An increased risk of adverse reactions and the required technical skills in the insertion of an infusion set are the main disadvantages [76]. Concerning the treatment of venomous animals’ bites/stings, the *i.v.* principle is the most recommended route for the administration of antivenoms at present and should be engaged whenever possible [83,84]. It can be performed by perfusion or by slow direct injection, with the latter becoming effective more rapidly and being less costly, also enabling urgent cessation upon the onset of immediate adverse reactions [85,86].

Antivenoms are large molecules whose absorption, when given by any route other than *i.v.*, occurs slowly via the lymphatics before their further distribution occurs [84]. Despite the complexity of the antivenoms’ pathway through the organism and the number of the involved body compartments (Figure 1), previous experimental studies (Table 1), performed with the aim of elucidating their pharmacokinetics, either alone or in combination with the respective venom, in the vast majority of cases, were limited to concentration level monitoring in the systemic circulation exclusively [43,87,88,89]. The main reason for the commonly used principle is self-explanatory concerning the sampling feasibility. However, in the frame of the venom/antivenom interplay, pronounced and easily traceable toxin-induced pathophysiological changes affecting the cardiovascular system as a whole, such as coagulation disorders, myoglobinuria and enzyme disturbances [90], also contribute to its widespread and deeply rooted application. In addition to the blood, there is a practice of antivenom quantity tracing over the time course in urine, as well as its detection in various organs, mostly to gain insight into the elimination process [43]. Over the years, not so recently, the need for expanding the research field to other relevant body compartments, primarily the lymphatic system, has been recognized, and it will be discussed later. In experimental investigations, among different available animal models, larger species, such as sheep [73], porcine [91,92], cattle [93] and especially rabbits [94], have been preferably used, enabling the extended sampling and supply of adequate amounts of testing material. Small animals, such as mice [95], rats [96] and guinea pigs [97], have been considered less useful, primarily because of their size, small muscle mass, poor physiological comparison with humans and, consequently, questionable translatability of the obtained cognitions to envenomed and/or treated patients [98]. Interestingly, in the past, dogs [99] and kittens [100] were also employed for antivenom pharmacokinetic studies but nowadays have been abandoned. Furthermore, the immune system of the above-mentioned species is different compared to the animals used for snake antivenom production: mostly horses and exceptionally sheep [101], donkeys and llamas [102]. Their antibodies are foreign to the animal model, which affects the maximum plasma concentration of the active compound and its elimination rate, as demonstrated in rabbits, mice, rats and cows [89,103,104], which should be kept in mind when comparing the results from studies performed on different species in which heterologous IgGs or their fragments exhibit inherent pharmacokinetic properties [105].

### 2.1. Antivenom’s Pharmacokinetic Profile in Animal Studies

According to the established opinion, *i.m.* administration results in antivenom’s slow and difficult appearance in the blood with the consequence of a long period required to achieve the maximum concentration (*t*_max_), poor bioavailability and the delayed or incomplete neutralization of the venom components [35]. As shown in rabbits, the absorption of the venom-specific antibodies is prolonged considering that the *t*_max_ in the blood varies between 48 and 76 h for IgGs and F(ab’)_2_ fragments [88]. The appearance of Fab fragments occurs faster with a *t*_max_ of around 12 h [43]. The bioavailability is low since only 36–42% of the total administered dose reaches the systemic circulation [87,89]. In the envenomation setup, venom components are usually much smaller and enter the bloodstream faster than antivenom applied by the *i.m.* route, which is why the general presumption about its inability to provide timely cessation of the toxins’ escape to the place of action was settled [35,87].

#### 2.1.1. Antivenom’s Impact on Venoms of Elapids and Scorpions

Most often, the pharmacokinetic behavior of the *i.m.* antivenom does not match that of the target venom. The discrepancy is particularly emphasized if envenomation is caused by venoms whose action is primarily neurotoxic and mediated by toxins of low molecular weight, such as those of scorpions and snakes from the Elapidae family [88,114]. In support of the rapidity of their absorption, there are observations indicating that sometimes they can become detectable in blood almost immediately, even in only a few minutes after envenomation [109,110]. Moreover, it has been proven that about 70% of the administered dose of *Leiurus quinquestriatus* venom enters the bloodstream within 15 min [115], 90% of *Walterinnesia aegyptia* venom within 60 min [103] and 96% of *Androctonus australis hector* venom within 30 min [113], showing almost complete absorption of the whole fraction from the injection site to the systemic circulation in a very short time. Furthermore, a *t*_max_ for scorpion venoms appeared to be less than 2 h [110,115], with the most common range between 30 and 60 min [110,111,116]. *Walterinnesia aegyptia* venom is believed to be among the ones that are characterized by exceptionally fast uptake since it reaches the maximum concentration within 5–20 min following *i.m.* injection [103]. Monitoring of the *Micrurus nigrocinctus* venom toxicokinetics also confirmed rapid absorption since detectable concentrations were measured within the first half hour after the inoculation [117]. A progressive increase in the circulating antigens’ level was observed, reaching a peak at approximately 2 h following the injection in rabbits and somewhat earlier in mice. Not only absorption but also distribution to peripheral compartments is considered to be a relatively fast process [118] with a half-life shorter than 30 min [109,116]. On the other hand, elimination from the body is usually measured within a greater time span [112,116].

The pharmacokinetic incompatibility between venoms injected either *i.m.* or *s.c.*, reflecting typical envenomation, and *i.m.* applied antivenoms was demonstrated by a number of rescue-type studies. According to Krifi et al. [109], it seems that complete neutralization occurs only after 7 h. The most probable reasons are associated with the limited bioavailability of antivenoms given *i.m.*, not exceeding 50% of the administered dose, but also a significantly longer *t*_max_, measured even two days post-treatment [87,89], indicating a considerable delay in the absorption process. It is a well-known fact that the release of high-molecular weight proteins from the *i.m.* or *s.c.* injection site occurs gradually [119], which is applicable to whole IgGs and their fragments as well. Hammoudi-Triki et al. [113] performed a toxicokinetic analysis of *Androctonus australis hector* venom in envenomed rats after their treatment with antivenom, either in the form of F(ab’)_2_ or Fab fragments. The F(ab’)_2_-based antivenom therapy by the *i.m.* injection neutralized toxins at a slower rate than the one carried out by the *i.v.* route. Moreover, the total amount of free venom absorbed in blood over a defined time frame was higher, and the extent of toxic fraction complexed with antibodies was lower. Comparable results were obtained when Fab fragments were employed, but the difference in the amount of bound venom between alternative injection routes was less pronounced.

Results obtained by monitoring the venom/antivenom levels in the systemic circulation suggest that *i.m.* antivenoms are not up to the task when effective neutralization of the lethal toxicity of scorpion and elapid venoms should be achieved [109]. Their pharmacokinetics does not act either temporally or quantitatively adapted to the significantly faster arrival of the respective venoms, whose toxins, due to their smaller size and greater diffusivity, appear in the blood much earlier than neutralizing antibodies. Given how quickly they are absorbed, distributed and eliminated, envenomation induced by venoms enriched with low-molecular weight peptides/proteins represents a life-threatening emergency and requires immediate attention [110]. Accordingly, an early *i.v.* injection of an appropriate antivenom dose is considered a more prosperous way for the achievement of rapid and permanent neutralization of circulating toxins [88]. Because of the venom’s large volume of distribution and the fact that antibodies are typically administered during its post-distributive phase, the probability of an antigen–antibody interaction is limited, so the antivenom’s efficacy mostly relies on its ability of forming immuno-complexes in the circulation, serving as a direct, immediate entry pathway for that given by the *i.v* route. Subsequent free venom level reduction promotes the redistribution of tissue-bound antigens from the extravascular space into the central compartment (blood) where their neutralization for the most part occurs [89,106,111,112]. The redistribution capability of F(ab’)_2_ fragments to alter venom’s pharmacokinetics is considered particularly suitable for use in the immunotherapy of scorpion and elapid bites [89,103,112]. It has been noticed that an elevation of the plasma venom level in the post-infusion period occurred and resulted in a 10- or even 76-fold higher area under the concentration–time curve in the F(ab’)_2_-treated group in comparison to the control, probably as a consequence of toxins’ redistribution and antibody-mediated sequestration [89,111].

Following *i.v.* administration, F(ab’)_2_-based antivenoms are usually fitted to a two- [89] or three-compartment open pharmacokinetic model [88,103], encompassing a central compartment (vascular system), a rapidly equilibrating shallow tissue compartment and a slowly equilibrating deep tissue compartment [35]. In comparison to whole IgGs, F(ab’)_2_ fragments not only possess a shorter *t*_max_ and distribution half-life in the circulation [103] but also, due to the larger volume of distribution, diffuse to the extravascular space to a greater extent, showing affinity to both shallow and deep tissue compartments where the target toxins subside [35]. Therapeutic appropriateness of *i.v.* administered F(ab’)_2_ fragments is supported by the finding that they require two- to three-fold less time to reach a *t*_max_ in the extravascular space [103]. Moreover, their mean distribution half-lives for the shallow and deep compartments are six and five times shorter, respectively [88]. On the other hand, it seems that *i.m.* administration diminishes the efficacy of F(ab’)_2_-based antivenoms, for now, still having little importance in the treatment of envenomation caused by scorpion and elapid bites [89,103], and only an early *i.v.* injection of an appropriate amount, preferably much higher than the minimum effective dose [103,106], can provide a fast and permanent neutralization of the circulating toxins. The observed difference between alternative routes has been straightforwardly demonstrated in the example of *Androctonus australis garzonii* venom that has been completely removed from blood in less than 10 min when specific antivenom was given *i.v.*, while it took even 8 h for its clearance when the same was applied *i.m.* [110].

#### 2.1.2. Antivenom’s Impact on Venoms of Viperids

On the contrary, Viperidae family venoms, in which higher-molecular weight proteins predominate, show different pharmacokinetic profiles [35]. In the beginning, distinctive fast absorption occurs since the venom components can be detected in blood already after 10–15 min [94,99], reaching a maximum concentration after several hours, as demonstrated in *Vipera aspis*-experimentally induced envenomation [94]. An initial phase of rapid absorption is followed by a prolonged period of gradual release from the subcutaneous tissue around the injection site into the circulation [120], lasting up to 24 h [121] or even 3 days [94], and is especially emphasized following *s.c.* administration of the venom when the extended elimination half-life lasting for up to 5 days was reported [122]. Not all viperid venoms’ uptake occurs to the same extent, with their bioavailability ranging from 4% [120] to 86% [123]. The fraction of injected components remains retained at the site of application functioning as a depot [124] and probably is responsible for the local tissue damage [125].

A delayed increase in the venom plasma concentrations may be associated with the absorption mediated by the lymph as well [62]. Specifically, following envenomation, venoms are delivered via the *s.c* or *i.m.* path into the interstitial space where they enter into the bloodstream either through blood or lymph capillaries [62]. The choice of transport is conditioned by the molecular weight of toxins and varies between small neurotoxins from the venoms of scorpions and snakes from the Elapidae family and larger haemotoxins from the venoms of snakes from the Viperidae family [126]. Direct access to the blood capillaries is possible only for peptides and proteins smaller than 9 kDa, while others (20–100 kDa) are mostly absorbed via the lymphatic system, which serves as a permanent source for their continuous delivery into the systemic circulation [127]. This is in accordance with the study of Porter et al. [128] who investigated *s.c.* administered therapeutic proteins and noticed that an increase in their size causes a reduction in the blood vascular endothelium’s permeability, redirecting the larger molecules toward the lymphatic system as an entrance for their uptake and further distribution. Nowadays, it is becoming more and more evident that the lymphatic system is also an important body compartment whose role in the neutralization process has been insufficiently investigated so far but could possibly provide new cognitions into a process of absorption and distribution [129]. Because of its low volume and relatively slow flow, the lymph should have an influence not only on the residence time in the body but also the absorption rate from the injection site to the bloodstream. Audebert et al. [94] showed that, although the whole venom fraction disappeared from the application site 7 h after *i.m.* injection, only 25% of the administered dose reached the vascular space, thus confirming the lymphatic system as the initial body compartment through which the absorption occurs, while release into the blood follows only afterward [43]. Moreover, the study in which the *Micrurus fulvius* envenomation progress was followed [73] unraveled that around 70% of the initial dose had been cumulatively absorbed by both compartments, of which even 25% via the lymphatic system. The results suggest that, together with the depot at the injection site, the lymph pool also provides a sustained inoculum of venom carried into the bloodstream, whose release can last for several days [73], resulting in the phenomenon of recurrent envenomation [74]. Because antivenom has a significantly higher clearance rate than some medically relevant toxins [124], local and systemic scenarios of worsening after initial improvement might occur. Briefly, Viperidae family toxins act in a more delayed manner, which emphasizes the relevance of the maintenance of high antibody levels in plasma long enough to assure repeated cycling through the interstitial fluid of organs as well as neutralization of venom components that may reach the circulation later on [35].

The efficacy of anti-viperid antivenoms given *i.m.*, just like that of antivenoms against scorpion and elapid bites, appears questionable on several grounds [107]. For instance, as clearly demonstrated in rabbits, their use is connected to a relatively poor bioavailability of 42% and slow absorption with a *t*_max_ of 48 h [87]. Additionally, *i.m.* injection may result in a large hematoma at the site of application, whose formation is associated with uncoagulable blood caused by viper envenomation [130]. Even though Fab fragments reach the bloodstream faster than whole IgGs and F(ab’)_2_ fragments, with a *t*_max_ of 12 h in rabbits [43], no improvement in the neutralization of *Bothrops asper* venom-induced lethality was noticed when neither of the three antivenom types was used [108]. Moreover, as observed by Riviere et al. [107], a delayed and only partial neutralization of *Vipera aspis* venom was achieved. A widely held belief that the *i.m.* route represents a poor method of antivenom administration was established decades ago and persisted ever since. Although resulting in incomplete uptake, a prolonged time to reach maximum concentration and a quite low *c*_max_, it may provide persistent plasma levels of antivenom that could be sufficient to prevent recurrent envenomation symptoms, especially coagulopathy, by maintaining a steady-state blood antibody concentration [131], probably on account of the extension of the apparent elimination half-life [87].

#### 2.1.3. Role of Lymphatic System in Venom Neutralization

Paniagua et al. [74] pointed out the importance of venom neutralization not only in the blood but also in the lymphatic system. In light of new cognitions, matching the venom/antivenom pharmacokinetics in the systemic circulation probably is not the only indicator of therapeutic effectiveness [125] since a critical part of the envenomation process and its containment must be played by lymph physiology as well [132], the impact of which has been largely neglected, as evident from the paucity of past research. *S.c.* venom absorption into the bloodstream, via the lymphatic system, was suggested as early as 1938 by Fidler et al. [132]. Three years later, Barnes and Trueta [100] demonstrated that snake venoms containing components of high molecular weight are not absorbed when lymphatic vessels are obstructed, contrary to those possessing smaller toxic molecules. By employing combined blood and lymphatic sampling in a central lymph cannulated sheep model, Paniagua et al. [73] made significant progress toward understanding how the venom passes from the site of injection into the systemic circulation. Their study proved that lymphatic absorption from subcutaneous tissue as the missing parameter plays a major role in its distribution and bioavailability. Namely, 25% of the absorbed dose was recovered via the lymphatic system. The highest concentration of venom found in lymph was more than 25-fold higher than that reaching the blood. In the following, most recent work, Paniagua et al. [74] enriched the study of antivenom pharmacokinetics in the systemic circulation by its simultaneous evaluation in the lymphatic system. From their work, which aimed at defining the role of lymphatic absorption in the neutralization of *s.c.* injected venom by the antivenom *i.v.* administered 2 h after envenomation, a few important discoveries emerged. First, antivenom can extravasate from the bloodstream into the lymphatic system, eliminating a substantial amount of venom (around 70%) before lymph reaches the systemic circulation. Second, in contrast with findings in the blood, where free venom dropped rapidly to undetectable levels following antivenom administration, an unbound fraction remained detectable in lymph until the end of the experiment. *I.v.* antivenom’s action in the lymphatic system, where it arrived by extravasation from the blood, seems to be slow and incomplete, probably because of its lower concentration than in serum. An alternative explanation might be that venom concentration exceeded that in serum due to absorption from the subcutaneous tissue at the injection site that acts as a persistent depot. Irrespectively, the rate of demonstrated lymph-phase neutralization is probably highly relevant for antivenom effectiveness, at least in the case of *i.v.* antivenoms, while the role and the impact factor of those given *i.m.* are yet to be investigated.

## 3. Clinical Studies

Clinical studies of antivenoms are generally performed with the objective of efficacy and safety assessment [133]. Despite the high importance of the latter, only summarized knowledge of the laboratory and clinical findings concerning their efficacy with a reference to the influence of the *i.v.* and *i.m.* routes on the treatment outcome was within this review’s scope. It is a well-known fact that the successful performance of antivenoms depends on their time and dose adjustment to the kinetics of the respective venom. Delivery mode represents one of the ways by which the harmonization of the venom/antivenom interplay leading to the neutralization and elimination of the pathophysiologically relevant toxins could be accomplished [35,37]. Namely, the optimal treatment protocol for snakebite management still remains controversial, mainly due to insufficient knowledge of the pharmacokinetics of venom and antivenom, as well as their interaction, limiting the evidence to support currently practiced administration principles [125]. Although the *i.m.* route is still sometimes practiced in the field [35,47], *i.v.* administration is the cornerstone principle for the antivenom application, probably because of strong recommendations from the authorities [52], grounded for the most part on conclusions from the numerous animal studies performed in an ideal experimental setup [134], on the basis of which insight into events in the systemic circulation was gained, as already discussed. However, antivenom pharmacokinetics appears to be species-dependent as a phenomenon that could possibly result in distorted predictions when translating the cognitions from animal models to humans [104,135].

Although highly needed, studies on healthy volunteers and envenomed patients (Table 2) are scarce and often flawed [47], providing insufficient data for unambiguous conclusions about the most efficient application strategy against snakebite envenoming [134]. In the vast majority of cases, they are performed in uncontrolled setting frequently including only individual cases [17,136,137,138] or groups small in the number of participants [45,46,124,139,140]. Often, there are situations where the species responsible for the envenomation could not have been reliably identified and the treatment could be suspected only from the patient’s description or the clinical signs, mostly coagulopathy as the most common one [2,45,46,134], which calls into question the appropriateness of the applied antivenom’s specificity and, consequently, the degree of its efficacy. Time elapsed between the snakebite incident and the therapy application usually varies between the individual cases, aggravating the comparison and interpretation of obtained results [45,94,138,140]. Finally, infrequent sampling during the first few hours after antivenom administration, with the majority of victims providing an inadequate number of time concentration samples [45], and an unsatisfactory long follow-up period, interrupted by the patient’s discharge from the hospital [140], represent the most common restricting factors for a proper pharmacokinetic analysis. Therefore, the accomplishment of the complete picture of administration route appropriateness aims at looking at the data from animal studies in a consolidated manner with those measured in treated patients, all in view of the course of the clinical progress.

### 3.1. Antivenom’s Pharmacokinetic Profile in Human Studies

#### 3.1.1. Pharmacokinetic Properties of *i.v.* Antivenoms

Various studies in humans have been performed with the aim to evaluate the pharmacokinetic properties of antivenoms in relation to the type of active compound they contain (IgGs, F(ab’)_2_ or Fab fragments) and the route of application [17,45,48,124,134,138,139,140]. The kinetics of *i.v.* administered antivenoms has been well described, revealing that in envenomed and treated patients, they follow a biphasic concentration decay pattern [35,45,134,136]. The initial rapid decline observed during the distribution phase is attributed to the formation of immuno-complexes with the venom components whose clearance from the circulation causes a concomitant and rapid decrease in toxins to undetectable levels within 5 min [45] to 60 min [46] upon the start of therapy. Due to its rapidity, the first phase could be easily missed and the distribution half-life miscalculated. A more prolonged decline that is characteristic of the terminal elimination phase reflects the clearance of the heterologous antivenom’s proteins from the central compartment by the reticuloendothelial system [35,45]. The elimination half-life of IgGs and F(ab’)_2_ fragments is relatively long so its accurate determination requires an extended, hardly achievable follow-up period. Thus, in comparison to many other drugs, the estimation of the pharmacokinetic parameters of antivenoms following *i.v.* administration might be inherently less reliable [45] which could be the reason for the observed quantitative variations between different studies in humans, although involving the same type of active compound regarding its molecular weight.

Most of the investigations were related to the antivenoms containing F(ab’)_2_ fragments. In one of the pioneering clinical studies, where antivenom for *Calloselasma rhodostoma* bite treatment was administered, the distribution half-life, determined on five patients, was only 0.3 h [45]. It appeared significantly shorter in comparison to the 4.6 or even 7 h determined for antivenoms against envenomation caused by *Daboia russelii* [134] or European vipers [17], respectively. Variations were observed between the elimination half-life values as well. The results of a clinical trial including 22 patients given equine F(ab’)_2_ antivenom (Ipser Africa) after *Echis ocellatus* envenomation demonstrated the elimination half-life of 18 h [48]. On the other hand, in another study on six participants treated with anti-*Vipera russelli* antivenom, it was twice as long [139]. Occasionally, even a more extended time needed for the removal of F(ab’)_2_ fragments was reported, ranging from 4 to almost 6 days [45,134]. The evaluation of systemic clearance in a single case report [17] and another study with five participants [45] revealed only slightly different values fluctuating between 1.6 and 1.7 mL/h/kg. The results regarding the volume of distribution were equally comparable, with values of 214 mL/kg [17] and 233 mL/kg [45].

In a comparative study that included antivenoms consisting of whole IgG molecules against *Calloselasma rhodostoma* bite, the differences between their pharmacokinetic parameters, calculated from 13 subjects, were also evident [45]. The distribution half-life of the preparation produced in goat was four times larger compared to that of the equine origin with median values of 2 h and 0.5 h, respectively. Almost twice as much time was needed for the elimination of equine (82 h) than goat IgGs (46 h), which expectedly influenced their clearance values (0.6 mL/h/kg vs. 1.3 mL/h/kg). The volume of distribution of around 90 mL/kg was the only parameter that proved consistent for both therapeutics. One case report of a patient given equine IgG antivenom provided an elimination half-life that appeared as long as almost 7 days following *i.v.* administration [137].

Concerning the antivenoms’ pharmacokinetic variability among independently performed studies, the parameters determined for those composed of Fab fragments were not an exception. Fab antivenom (EchiTab), used in a clinical trial on 17 victims envenomed by *Echis ocellatus*, had a mean elimination half-life of around 4.3 h [48]. A significantly higher value was reported for antivenom against *Vipera berus* envenomation (ViperaTAb) whose Fab level decreased with an elimination half-life of 24 h on average (nine patients, range 9–50 h) [140], and which was in line with the values from three consecutive case series reports, spanning from 14 to 56 h [138]. Plasma concentration of Fab antivenom in four victims of crotaline envenomation needed 18 h to be reduced by half [124], similar to that of Sri Lankan Russell’s viper venom-specific antivenom (PolongaTAb), which was administered to 33 patients [46]. Its elimination half-life varied between 16 and 28 h. Regarding the volume of distribution, Seifert et al. [124] demonstrated that antivenom used against crotaline bite had a value of 110 mL/kg. According to two case series reports [138,140] that were related to the treatment of European vipers with ViperaTAb, the volume of distribution could be as large as 182–415 mL/kg and 118–524 mL/kg, respectively. The obtained results for the distribution half-life and systemic clearance appeared to be more uniform with the values in the range from 1.2 to 3.2 h [124,138] for the former and 4.3 to 13.4 mL/h/kg for the latter [124,138,140].

#### 3.1.2. Pharmacokinetic Properties of *i.m.* Antivenoms

Research providing a detailed pharmacokinetic profile of *i.m.* antivenoms is poor. Vázquez et al. evaluated the kinetics of scorpion antivenom on healthy volunteers. In one study, it was given by the *i.m.* (six subjects) [154] and in another by the *i.v.* route (eight subjects) [153]. When administered *i.m.*, there was no more than 17% of the antivenom content detectable in plasma at any time. The period needed for reaching its maximum concentration was 45 h, while after an *i.v.* bolus, the peak occurred in less than 5 min. The mean residence time was three-fold longer for *i.m.* than for *i.v.* antivenom. Equally so, the two routes differed in other pharmacokinetic parameters which additionally reinforced the opinion about *i.m.* administration as inferior, leading to the recommendation that it should not be practiced. In a prospective study comprising snake victims envenomed by *Vipera ammodytes*, a comparison of the pharmacokinetic profile of *i.v. Vipera berus* venom-specific Fab fragments (ViperaTAb) and *i.m. Vipera ammodytes* venom-specific F(ab’)_2_ fragments (European viper venom antiserum) was performed (nine subjects) [140]. Fab antivenom, due to the smaller size of its active compound, had a 2.5 larger volume of distribution and, since being given *i.v.*, reached maximum concentration in blood within 2 h. F(ab’)_2_ antivenom was gradually released from the muscle tissue into the systemic circulation. Its level peak occurred after only 70 h on average. On the other hand, F(ab’)_2_ antivenom had 25-fold longer total body clearance and a 14-fold longer elimination half-life compared to that administered *i.v.* (2 weeks vs. 24 h, respectively). The kinetics of Fab fragments after one or more *i.v.* applications matched better with the venom concentration in the early phase of envenomation compared to F(ab’)_2_ fragments that were given *i.m.* only on admission. *I.m.* use of F(ab’)_2_ fragments resulted in a slow rise of antivenom serum concentration that demanded their early administration but without the need for additional doses for the complete resolution of all clinical signs. *I.v.* use of Fab fragments resulted in an immediate rise in antivenom serum concentration that enabled their use according to the clinical progress, but it required multiple doses for an efficient therapy outcome.

### 3.2. Clinical Outcome

Venomous snakes belonging to either the Elapidae or Viperidae family are known to bring about a wide range of physiological disturbances [61,157]. The elapid venoms comprise toxins affecting the nervous system. They are also associated with numerous other serious systemic effects, while local tissue damage is minimal, with the exception of some *Micrurus* species [158]. The viperid venoms, besides the venom of *Crotalus durissus terrificus*, only occasionally cause neurotoxic signs. They act mainly on blood coagulation and induce strong necrosis at the bite site. Although it is obvious that many measurable diseases can be considered as relevant markers of antivenom’s efficiency depending on the route of application, in this review, a decision was made to put an emphasis only on venom-induced consumptive coagulopathy as the most common medical condition which is mutual to both elapids and viperids [2,7,159], primarily for the purpose of easier follow-up. The majority of clinical studies and individual case reports are related to *i.v.* administration and its successfulness in the antivenom-mediated reversal of the envenomation signs and symptoms. Those involving the *i.m.* route are much less represented, emphasizing a need for filling the gap. Moreover, there are no studies comparing *i.v.* and *i.m.* administration principles.

#### 3.2.1. Clinical Outcome after *i.v.* Antivenom Administration

As shown by a randomized, double-blind comparative trial of three IgG- or F(ab’)_2_-based antivenoms performed with the aim of assessing their efficacy and safety in the treatment of crotaline snakebite, all were capable of permanently restoring blood coagulability at 6 h and 24 h after the initial dose application in the great majority of investigated cases [146]. Specific antibodies persisted in the serum for at least a 48 h-long period, following which venom antigens became undetectable. Similarly, another trial demonstrated comparable effectiveness of two IgG antivenoms which permanently restored blood coagulopathy indicative of systemic envenomation by *Echis ocellatus* also at 6 h after the treatment but in a slightly smaller percentage of the participants [14]. The time span from antivenom *i.v.* administration to the normalization of hemostatic disturbances varied between different examinations, but generally, it can be noted that resolution within 24 h occurred. Equine F(ab’)_2_ antivenom against envenomation caused by European vipers (Viperfav) reversed the recurrence of coagulopathy symptoms immediately after the repeated application [17]. Timewise, Viperfav was equally successful in normalizing blood coagulation disorders associated with *Vipera berus* and *Vipera aspis* snakebites following the use of only one dose, with no recurrence of clinical or laboratory abnormalities [18]. FAV-Africa antivenom, also containing F(ab’)_2_ fragments, resolved hemorrhage in a day [147]. With Viperfav and African Antivipmyn antivenoms, improvement took place after just 6–12 h [148] or even 2–4 h [133], respectively.

Ovine Fab-based antivenoms have been used to treat systemic envenoming caused by European adders [51,160,161], North American crotalids [49,141] and carpet vipers [48,162]. They have the largest distribution volume of all due to small-sized active compounds that penetrate rapidly into the extravascular space where they enable prompt neutralization. However, Fab fragments are short-lived, and due to their premature elimination and insufficient plasma concentration, by the time late venom absorption from the depot at the site of inoculation occurs [37,124], the reappearance of envenomation follows frequently, as reported in many clinical studies [46,48,49,163], and much more often than for the other two types of antivenoms [14,45,133,143,164]. The pharmacokinetic analysis of ViperaTAb, an antivenom employed in a prospective case study of patients bitten by *Vipera ammodytes*, revealed that Fab fragments induced immediate venom level decrement, although lasting only temporarily [138]. A few hours later, patients again developed profound thrombocytopenia that was in correlation with the venom reappearance in the circulation. A similar observation resulted from a preliminary dose-finding study of patients treated with *Daboia russelli*-specific antivenom [46]. If an initial dose was too low to produce circulating levels of antivenom that can persist for long enough to cover continuing absorption of venom, durably abolishing its antigenemia, in the majority of participants, permanent restoration of blood coagulability and cessation of systemic bleeding could not be achieved. Equally, CroTAb antivenom initially improved the local manifestations of pit viper envenomation, but more than half of the patients enrolled in the clinical trial developed late, persistent or recurrent coagulation abnormalities that lasted for up to 2 weeks [49]. Ruha et al. [141] also reported only a transient advance of clinical signs in patients receiving CroFab antivenom which effectively controlled the consequences of rattlesnake envenomation at initial check points, but on follow-up, the subsequent appearance of delayed-onset coagulopathy and severe thrombocytopenia emerged. It has been concluded that the kinetics of *i.v.* administered Fab antivenoms probably matches better with the venom concentration in the early phase of envenomation, but for a complete improvement, multiple-dose administration might be needed [46,138].

Clinical implications of an inadequately long plasma persistence of Fab fragments appeared especially prominent in trials performed with the aim of their comparison with F(ab’)_2_ antivenoms. Ariaratnam et al. [142] suggested that a single dose of PolongaTAb, which was supposed to replace ineffective and unsafe F(ab’)_2_ antivenom against Russell’s viper bite, permanently restored blood coagulability in less than half of the patients, while maintenance dosing was required for the rest. On the other hand, in the F(ab’)_2_ group, the majority of enrolled subjects had restored coagulability after just one *i.v.* antivenom application, also showing a tendency toward a more rapid resolution of other systemic manifestations. Their results are consistent with other clinical trials. Boels et al. [143] reported a greater requirement for maintenance dosing and a higher incidence of symptom worsening in the Fab group over the F(ab’)_2_ group. Moreover, when comparing late coagulopathy in snakebite patients treated either with F(ab’)_2_ or Fab antivenom, Bush et al. [144] concluded that the former one significantly reduced late subacute coagulopathy without the need for additional doses, while the Fab-treated group was at an increased risk of the delayed onset of serious bleeding complications associated with recurring venom antigenemia and an accompanying drop in platelet count and fibrinogen levels. With regards to the efficacy of F(ab’)_2_ and Fab antivenoms, results from the study of Boyer et al. [145] clearly indicated that, regardless of which IgG derivative is used, a swift response to *i.v.* treatment evidenced by the normalization of the coagulation parameters can be expected but only during the acute phase of envenomation. When Fab antivenom is cleared, the ongoing presence of venom may result in delayed or recurrent coagulopathy.

#### 3.2.2. Clinical Outcome after *i.m.* Antivenom Administration

For now, there is not enough research being conducted that deals with the question of *i.m.* antivenom administration, especially when it comes to the straightforward comparison of its efficacy with the *i.v.* principle. One of the earliest studies reported that the correction of the *Ancistrodon rhodostoma* venom-induced coagulation defect occurred on average in 18 h (range 12–36 h) following the *i.m.* injection of the specific antivenom, which might be rather slow since the improvement was observed twice as fast (range 2–18 h) when *i.v.* application was employed [149]. However, it is important to emphasize that, when considering the antivenom efficacy in light of the administration route, the time elapsed from the incident to the treatment onset should be considered as well since it represents another factor with an important implication on the therapy outcome [153]. Late arrival to the hospital leading to a delay in antivenom application is the main determinant of poor prognosis as it bears the risk of severe envenoming symptom development with potentially fatal consequences [39,42,55,137,165,166]. Keeping in mind that snakebite incidents mostly happen in distant rural areas, far from medical health centers, *i.m.* administration still represents well-justified pre-hospital first aid, despite its proven unfavorable pharmacokinetics during the early phase of envenomation. As shown by Win-Aung [165], patients bitten by Russell’s viper who received *i.m.* antivenom in the field, within 2 h after the incident and prior to standard *i.v.* therapy, had their blood venom level reduced by more than half at the time of admission to the hospital when compared to the victims that were not treated until hospitalization, indicating its contribution to the neutralization of circulating toxins. As a consequence, the number of patients who developed systemic clinical and biochemical disorders was reduced and so was the fatality rate. One of the antivenoms whose *i.m.* administration has been implemented into practice in accordance with the national guidelines is *Vipera ammodytes ammodytes* venom-specific antivenom (European viper venom antiserum, in the literature also known as Zagreb antivenom, Institute of Immunology Inc., Zagreb, Croatia). It is clinically successful against homologous venom, as well as against the venoms of several other medically important European snakes, as demonstrated by its continuous, over-30-year-long use for the treatment of envenomings induced by *Vipera aspis* (Italy), *Vipera berus* (UK, Sweden), *Macrovipera lebetina* and *Montivipera xanthina* (Greece, Turkey) [167], interrupted in 2015 due to manufacturing discontinuation. In retrospective studies, more than 500 adults [19] and 160 children [150], presenting for the most part a mild to severe clinical picture, were analyzed. Almost all subjects received immunotherapy (99.7%). Their complete recovery was reported, since the withdrawal of all symptoms and signs of envenomation, which were mainly a result of the venom’s hematotoxic effects, occurred during the hospital stay. Only one case of a child bitten directly on the neck was fatal. Lukšić et al. [90] presented two clinical cases of moderate or severe impairment due to envenomation by *Vipera a. ammodytes* venom. *I.m.* administration of antivenom resulted in rapid improvement. Severe coagulopathy with the occurrence of profound thrombocytopenia resolved in less than 3 h, even when the therapy was applied with a significant delay of 16 h post-bite. Recently, a prospective study of *Vipera a. ammodytes*-envenomed patients that were treated *i.v.* with paraspecific ViperaTAb or *i.m.* with Zagreb antivenom with the aim of a comparison of their clinical efficacy was described [140]. It was the first one to examine the consequences of two different practices used in the treatment of victims who, by chance, had similar venom concentrations, as well as symptoms and signs of envenomation on admission before the antivenom was given. It was demonstrated that both therapies were statistically equally effective, since the outcomes, including the survival and length of the hospital stay, did not differ between the groups. Irrespective of the employed administration principle, the development of all medically significant complications was prevented, including further progress of thrombocytopenia that was effectively reversed. The only exception was neurotoxicity for which ViperaTAb proved to be ineffective due to the lack of specific antibodies. Apart from Zagreb antivenom, there is clinical evidence, although very limited, for some other European antivenoms which demonstrate effectiveness after *i.m.* administration [151,152]. The duration of the hospital stay, as another reasonable marker of antivenom effectiveness, was shorter for patients pre-treated with *i.m.* antivenom compared to those receiving only *i.v.* therapy (6 vs. 8 days) [165]. Two large retrospective clinical studies employing only the *i.m.* route for the application of antivenom against *Vipera ammodytes* venom showed that the average time of hospitalization was 3–13 days depending on the severity of the envenomation [19,150]. There are few clinical studies describing a similar span of hospital stay when antivenom was given *i.v.* [48,134,141], although some exceptions can be found. Chippaux et al. [147] reported that the mean time of hospitalization was 6.6 days, but it seems that its duration can be even shorter, ranging between 1 and 5 days [141,142,143,148,168].

Snakebites are rarely treated by *i.m.* antivenoms. So, most of the knowledge gained so far comes from research on antivenoms against venomous spiders [169,170] and scorpions [39] that are commonly administered by the *i.m.* route as it is considered safer, with a lower probability of inducing immediate-type hypersensitivity reactions [60]. However, the results regarding their effectiveness depending on the administration principle are still quite contradictory. According to the report on four cases of severe red-back spider envenomation, there was none or minimal response to treatment with *i.m.* applied antivenom, while the subsequent *i.v.* injection of an additional dose proved to be highly effective resulting in an almost complete resolution of all symptoms within 4–8 h [59]. Similarly, in a clinical trial of the efficacy and safety of new equine F(ab’)_2_ antivenom in the treatment of latrodectism, the achievement and maintenance of a clinically significant reduction in pain for 48 h post-treatment in the *i.v.*-treated group compared to placebo was recorded [155]. On the contrary, Isbister et al. [60] found the differences between the *i.m.* and *i.v.* routes insufficient to justify choosing one over the other after a clinical trial was performed on more than a hundred patients with moderate to severe latrodectism. Both principles were similarly efficient in reducing pain 2 h after the treatment. The *i.m.* group was more likely to benefit from improved systemic effects, while *i.v.* antivenom-treated participants were less likely to need additional doses and more likely to have improved pain 24 h post-therapy. The results related to the primary outcome of another comparative trial were in favor of *i.m.* antivenom as its application significantly reduced pain in red-back spider victims already at 1 h after the treatment, which could not be accomplished when the *i.v.* route was employed [156]. At 24 h, as a secondary outcome, the clinical picture of the *i.m.* group showed no improvement which was interpreted by antivenom’s delayed absorption and partial bioavailability. The *i.v.* group was significantly better. Ghalim et al. [39] also demonstrated the prompt efficiency of *i.m.* antivenom in counteracting scorpion envenoming signs, which was accompanied by a drop in venom blood concentration in comparison to the untreated group. In addition, a significant alleviation of local symptoms was observed 3 h following the therapy. However, a pharmacokinetic analysis of antivenom against widow spider bite revealed that, when the *i.m.* route was employed, it remained undetectable in the blood for at least 5 h post-therapy, while measurable concentrations in the systemic circulation were achieved already 30 min after completing the *i.v.* infusion [38]. The results agree with those of Krifi et al. [42] who reported successful and rapid clearance of scorpion venom from the blood following *i.v.* administration of antivenom, while that given *i.m.* failed to produce a significant effect on the toxicokinetic curve since the venom plasma concentration decreased over the next 6 h at a rate almost identical to the one observed among untreated victims. Complete elimination of toxins from the blood was achieved only when an additional *i.v.* dose was given.

## 4. Conclusions

With regards to the administration route of antivenoms against envenoming caused by snakes, but also spiders and scorpions, there is no unique practice in human therapy. Although clinical data are insufficient, a recommendation that antivenoms should preferentially be administered *i.v.*, as a principle of harmonizing their pharmacokinetics to that of the target venom, was introduced since it should eliminate the restraint associated with the *i.m.* route. Eventually, it got primacy among authorities. In spite of that, antivenoms given *i.m.* are also used in the field. The scientific explanation for the discrepancy between the proposed inferiority of *i.m.* administration in comparison to that performed *i.v.* and their comparable effectiveness is yet to be found. It seems that the venom neutralization in the lymphatics may be of importance for the clinical outcome, at least when the *i.v.* route is applied. The role of *i.m.* administered antivenom in the elimination of lymph-absorbed venom might be even greater, but it has not been studied yet. In other words, the matching of antivenom and venom appearance in blood might not be the only indicator of treatment success. Lower bioavailability associated with *i.m.* administration might be of lesser importance as well, considering that antivenom could provide substantial neutralization activity in the lymphatic system, eliminating venom before it reaches the bloodstream. For now, an unambiguous conclusion about the more effective route of antivenom administration still cannot be drawn. In an ideal scenario, both therapeutic principles should be compared in a comprehensive preclinical study involving IgG, F(ab’)_2_ and Fab antivenoms of identical specificity and potency, using the same model, and evaluating their pharmacokinetics on experimentally envenomed as well as on healthy animals, preferably in all relevant body compartments in which antivenom-mediated neutralization occurs.

## Figures and Tables

**Figure 1 toxins-15-00398-f001:**
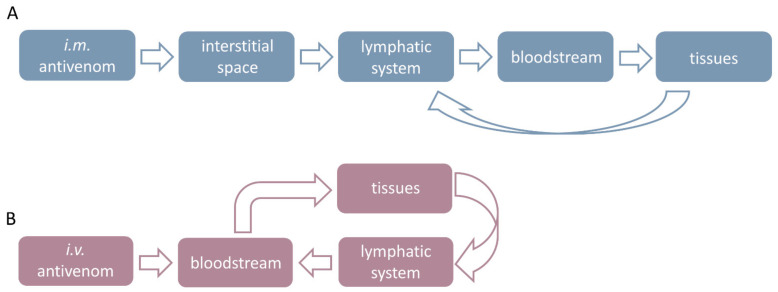
Distribution of *i.m.* (**A**) and *i.v.* antivenom (**B**) through the body compartments.

**Table 1 toxins-15-00398-t001:** Preclinical studies of antivenoms administered by *i.v.* or *i.m.* route measured in animal models.

Venomous Species	Route	Type of Antivenom	Animal Model							
			Rabbit	Sheep	Mouse	Porcine	Cattle	Horse	Rat	Dog
**Snakes**	*i.v.*	F(ab’)_2_	[43,87,88,106,107]	[74]	[95]					
		Fab	[43,107]			[91]				
		IgG	[88],		[95]		[93,104]	[104]		[99]
	*i.m.*	F(ab’)_2_	[87,88,107]		[95]					
		Fab	[43]		[108]					
		IgG	[88]		[95,108]					
	*s.c.*	Fab				[91]				
**Scorpions**	*i.v.*	F(ab’)_2_	[89,103,109,110,111,112]						[113]	
		Fab	[103]						[113]	
		IgG	[103]							
	*i.m.*	F(ab’)_2_	[89,109,110,112]						[113]	
		Fab	[103]						[113]	

**Table 2 toxins-15-00398-t002:** Clinical studies of antivenoms administered by *i.v.* or *i.m.* route.

Venomous Species	Route	Type of Antivenom	References
**Snakes**	*i.v.*	Fab	[46,48,49,124,138,140,141,142,143,144,145]
		F(ab’)_2_	[17,18,45,48,133,134,139,142,143,144,145,146,147,148]
		IgG	[14,45,137,146,149]
	*i.m.*	F(ab’)_2_	[19,90,140,150]
		IgG	[149,151,152]
**Scorpions**	*i.v.*	F(ab’)_2_	[42,153]
	*i.m.*	F(ab’)_2_	[39,42,154]
**Spiders**	*i.v.*	IgG/F(ab’)_2_	[38,59,60,155,156]
	*i.m.*	IgG/F(ab’)_2_	[38,59,60,156]

## Data Availability

Not applicable.

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
