# Peer review of "Snake Antivenoms—Toward Better Understanding of the Administration Route"

_toxins, 2023, doi:10.3390/toxins15060398_

Round 1

Reviewer 2 Report

The current study sumarrized the elimination of venom toxins from systemic circulation and lymphatic system on the respect of pharmacokinetics and toxicokinetics, it would be help to improve the cognition of the venom-antivenom interaction and optimize the antivenom application.

Before accept this study, I suggest the authors to add an overview flowchart about elimination routes, features, rate, applied object and administration prevalence, and also conduct some other small revision.

1. The "Key Contribution" should be improved, at least delete the advance "Additional......needed".

2. line47: delete "only".

3. line49: shortage of antivenoms.

4. line135-137, should be supported by relevant references.

5. The family name-"Elapidae" and "Viperidae" do not need italic.

6. In the conclusion, may the authors can propose a more detailed and feasible strategy for scientific study or clinical practice of elimination of venom toxins from systemic circulation and lymphatic system.

7. Whether the small molecular drug (Fab) would be more suitable for i.m. administration, and it should be discussed.

8. The progenostic effect of i.m. administration should be discussed.

9. References: species name should be wrote in italic.

Reviewer 3 Report

The author(s) has written an excellent and very interesting paper providing a review of the pharmacokinetics and toxicokinetics regarding the pathogenesis of snakebite envenomation and its treatment.  The paper is organized and covers the topics thoroughly. The references are ample and accurately used. I found only a few areas requiring English corrections. 

I did not find any areas in need of re-write.

This paper is very well written, except for a few areas requiring re-wording. These are indicated in the attached PDF.
